# Ultrasound Landmarks in the Approach to the Common Peroneal Nerve in a Sheep Model—Application in Peripheral Nerve Regeneration

**DOI:** 10.3390/life13091919

**Published:** 2023-09-15

**Authors:** Rui Alvites, Bruna Lopes, Patrícia Sousa, Ana Catarina Sousa, André Coelho, Alícia Moreira, Alexandra Rêma, Luís Atayde, Carla Mendonça, Ana Lúcia Luís, Ana Colette Maurício

**Affiliations:** 1Departamento de Clínicas Veterinárias, Instituto de Ciências Biomédicas de Abel Salazar (ICBAS), Universidade do Porto (UP), Rua de Jorge Viterbo Ferreira, No. 228, 4050-313 Porto, Portugal; ruialvites@hotmail.com (R.A.); brunisabel95@gmail.com (B.L.); pfrfs_10@hotmail.com (P.S.); anacatarinasoaressousa@hotmail.com (A.C.S.); andrefmc17@gmail.com (A.C.); alicia.moreira.1998@gmail.com (A.M.); alexandra.rema@gmail.com (A.R.); ataydelm@gmail.com (L.A.); cmmendonca@icbas.up.pt (C.M.); alluis@icbas.up.pt (A.L.L.); 2Centro de Estudos de Ciência Animal (CECA), Instituto de Ciências, Tecnologias e Agroambiente da Universidade do Porto (ICETA), Rua D. Manuel II, Apartado 55142, 4051-401 Porto, Portugal; 3Associate Laboratory for Animal and Veterinary Science (AL4AnimalS), 1300-477 Lisboa, Portugal; 4Instituto Universitário de Ciências da Saúde (IUCS), Cooperativa de Ensino Superior Politécnico e Universitário (CESPU), Avenida Central de Gandra 1317, 4585-116 Gandra, Portugal

**Keywords:** peripheral nerve, peroneal common nerve, ultrasonography, ultrasound landmarks sheep model

## Abstract

Peripheral nerve injury (PNI) remains a medical challenge with no easy resolution. Over the last few decades, significant advances have been achieved in promoting peripheral nerve regeneration, and new assessment tools have been developed, both functional and imaging, to quantify the proportion and quality of nerve recovery. The exploration of new animal models, larger, more complex, and with more similarities to humans, has made it possible to reduce the gap between the results obtained in classic animal models, such as rodents, and the application of new therapies in humans and animals of clinical interest. Ultrasonography is an imaging technique recurrently used in clinical practice to assess the peripheral nerves, allowing for its anatomical and topographic characterization, aiding in the administration of anesthesia, and in the performance of nerve blocks. The use of this technique in animal models is scarce, but it could be a useful tool in monitoring the progression of nerve regeneration after the induction of controlled experimental lesions. In this work, sheep, a promising animal model in the area of peripheral nerve regeneration, were subjected to an ultrasonographic study of three peripheral nerves of the hind limb, the common peroneal, and tibial and sciatic nerves. The main aim was to establish values of dimensions and ultrasound appearance in healthy nerves and landmarks for their identification, as well as to perform an ultrasound evaluation of the cranial tibial muscle, an effector muscle of the common peroneal nerve, and to establish normal values for its ultrasound appearance and dimensions. The results obtained will allow the use of these data as control values in future work exploring new therapeutic options for nerve regeneration after induction of common peroneal nerve injuries in sheep.

## 1. Introduction

Peripheral nerve injuries (PNI) are common occurrences in both humans and animals with severe physiological and functional consequences [1]. Despite recent advances in promoting peripheral nerve regeneration after injury, traditional surgical methods continue to be the gold standard treatments commonly used, and therapeutic alternatives with equal effectiveness have not yet been established [2].

The overwhelming majority of works in which new therapeutic alternatives for the treatment of PNI are studied are based on more traditional animal models, such as rodents [3], in which significant advances have been achieved in the establishment of new innovative treatments, namely in the fields of regenerative medicine and based on cell-based therapies and application of biomaterials [4]. For instance, previously, our research group has developed several new therapeutic combinations based on the use of innovative biomaterials and mesenchymal stem cell-derived products, which have demonstrated good results when applied in the rat model [5,6].

Despite the undeniable advances achieved in recent years, the use of a more complex animal model, in a translational and scale-up perspective before application in real clinical scenarios in veterinary and human medicine, is still limited by several challenges and has faced some resistance in the scientific community dedicated to this field of research [3]. Among the options for more complex animal models with several advantages to be used in in vivo PNI works is the sheep, a species with dimensional, anatomical, and neurophysiological characteristics very similar to humans and which, furthermore, presents a whole range of technical advantages such as its low acquiring and maintenance costs, tendentially docile temperament and few ethical restrictions associated with its scientific use [7]. In fact, recently, this animal model has begun to gain some attention in this field of research [8]. In the past, our group developed a new protocol for the induction of lesions in the common peroneal nerve of a sheep model and a set of functional assessment techniques to monitor nerve recovery after injury [9]. In this work, where a detailed anatomical description of the sciatic nerve and its branches in sheep was also made, it was possible to establish standardized protocols for the induction of neurotmesis injuries in the peroneal common nerve to test the application of therapeutic options, such as tube guides and end-to-end sutures, to develop functional assessment methods such as a withdrawal reflex, proprioceptive assessment and gait characterization, and also to obtain default stereological values for the intervened nerves and for healthy ones, which can now be used as control values in new projects. 

In a peripheral nerve regeneration assay is important for the evaluation of regeneration to be multimodal and based on a multiapproach scheme, founded both on the determination of motor and sensory functional recovery as well as on the imaging and histomorphometric evaluation of the intervened nerves and their effector organs [10,11]. Only in this way can it be confirmed whether regeneration translates into recovery of nerve form and function. Contrary to what happens in rodents, where reduced dimensions can be a limitation to the use of a greater variety of certain techniques, the dimensions of the ovine model allow the application of diagnostic tools closer to those used in companion animals and humans, such as the evaluation of nerve conductivity and ultrasonography, techniques whose use in this field of research are still not systematic. 

Specifically, ultrasonography is an imaging technique used to assess the peripheral nerves of human patients. It allows them not only to carry out their morphological and isolated assessment regarding the neighboring tissues but also to be used as a guiding technique during local anesthetic blockages [12,13]. This technique also allows the identification of some pathologies of the peripheral nerve, such as compression injuries, perineural fibrosis, inflammation, demyelination, and the presence of tumors, making it possible to monitor the progression of the nerve injury or its resolution [14]. The authors believe that in studies using large animal models such as sheep, ultrasonography could be an additional and effective tool that can allow monitoring the progression of the site of the induced nerve injury after the application of a given therapeutic approach, thus allowing, for example, to monitor the growth of nerve tops into an induced nerve gap and the degree of fibrosis and edema around the nerve under regeneration.

Most studies of ultrasonographic description of peripheral nerves in veterinary medicine were carried out in small animals such as dogs and cats [14,15,16]. In this work, an ultrasound evaluation of ten sheep’s sciatic nerves and their main branches, the common peroneal nerve and the tibial nerve, was carried out in order to establish the ultrasound appearance of these structures in non-intervened nerves and their anatomical and regional relationship to neighboring structures. Likewise, the *tibialis cranialis* muscle, an effector muscle of the common peroneal nerve, was also evaluated in terms of its ultrasound characteristics and dimensions. Subsequently, the registered features may be used as reference values in comparative studies with peroneal common nerves subjected to controlled injuries and with cranial tibial muscles that have suffered a consequent atrophy due to denervation, maximizing the number of parameters to be evaluated in the intervened animals and ensuring a better determination of the performance of the applied therapeutic options. The main objective is to establish an ultrasound approach technique and standard values that can be used as references in future assays on peripheral nerve regeneration in the sheep model. 

## 2. Materials and Methods

### 2.1. Animals 

All procedures performed on animals were approved by the Organism Responsible for Animal Welfare (ORBEA) of the Abel Salazar Institute for Biomedical Sciences (ICBAS) from the University of Porto (UP) (project 459/2023/ORBEA) and by the Veterinary Authorities of Portugal (DGAV) (project DGAV: 2018-07-11014510), taking place in facilities previously approved by the official authorities (Clinical and Veterinary Research Center of Vairão—CCIVV). All activities were carried out following the assumptions present in the Portuguese decree law DL 113/2013, adapted from the EU directive 2010/63/EU of the European Parliament, and the OECD Guidance Document on the Recognition, Assessment and Use of Clinical Signs as Humane Endpoints for Experimental Animals Used in Safety Evaluation (2000). Additionally, whenever possible, all measures were taken to avoid and minimize any event of discomfort or pain in accordance with the humane endpoints for animal suffering and distress. The animals were not subjected to any type of surgical intervention or injury induction before the ultrasound scans were performed and were reused from other scientific studies in which there was no relationship or interference with the characteristics of the peripheral nerve, thus ensuring compliance with the assumptions of animals reusing for scientific purposes.

Ten sheep (*Ovis aries*), merino breed, female gender, 5 to 6 years, and 50–60 kg BW were used in this characterization work. All animals were purchased from authorized national producers previously approved by the host institution, officially brucellosis-free, and subjected to premovement testing for infectious diseases. After the reception, the animals underwent a complete clinical evaluation, and a prophylaxis protocol was instituted, including internal deworming, vaccination against enterotoxemia, and corrective trimming of the hooves. Regularly throughout the stabling period and immediately prior to the ultrasound scans being performed, the animals were subjected to complete physical examinations and assessments of their general well-being. The sheep were kept stabled in community groups in order to guarantee the expression of their gregarious behavior. They were fed concentrate and hay in a concentration and frequency adapted to their nutritional needs and had permanent access to fresh water.

To perform the ultrasound scans, the animals were sedated with xylazine (Rampun^®^, Bayer, Leverkusen, Germany, 0.1 mg/Kg, IM) and butorphanol (Alvagesic^®^, HiFarmaX, São Domingo de Rana, Portugal 0.05 mg/Kg, IM). Dosage reinforcements were applied when necessary to ensure well-being and comfort throughout the examination. After the procedure, the sheep were allowed to recover naturally or through the administration of atipamezole hydrochloride (Antisedan^®^, Zoetis, Parsippany, NJ, USA, 0.025 mg/Kg IM) when necessary. 

### 2.2. Ultrasound

#### 2.2.1. Preparation of the Ultrasound Field

After sedation, the animals were placed in lateral decubitus on a surgery table, followed by the preparation of the ultrasound field to be evaluated. The hind limb was trichotomized on its lateral side from the gluteal area to the talocrural joint and then cleaned with chlorhexidine. The same preparation was performed bilaterally since the ultrasound evaluation of nerves and muscles was performed sequentially on both limbs. Acoustic gel was used to improve image acquisition.

#### 2.2.2. Ultrasound Nerve Evaluation

Ultrasound scans were performed using the MyLab™ VET ultrasound scanner equipped with an SL1543 linear probe (4–13 MHz, 47 mm) (Esaote^®^, Genoa, Italy). The space between the greater trochanter of the femur and the ischial tuberosity was used as a reference point for the site of passage of the sciatic nerve after its emergence from the greater sciatic foramen, with ultrasound exploration starting distally, in the middle of the lateral surface of the thigh (Figure 1a). The probe was applied over the skin and *fascia lata* in the intermuscular groove between the *vastus lateralis* and *biceps femoris* muscles, parallel to the longitudinal axis of the hind limb, slightly inclined cranially (Figure 1b). Then, the probe was slowly displaced in a distal–proximal direction in order to allow the identification of the sciatic nerve after its passage between the reference bone prominences. Next, the sciatic nerve was followed distally and longitudinally to the site of emission of its two main branches, common peroneal and tibial nerves, proximally to the stifle. The two derived nerves were also followed distally up to the level where the deep position of both made it impossible to continue the lateral ultrasound progression. During the ultrasound exploration, the diameters of the sciatic nerve were measured in the middle of its course along the thigh and immediately before its division. The diameters of the common peroneal and tibial nerves were also determined immediately after its emission. Each site was measured three times on each animal to minimize the effect of variations in the probe position and in the measurement site. The ultrasound characteristics of each nerve and its relationship with neighboring soft tissues were also evaluated. 

#### 2.2.3. Ultrasound Muscle Evaluation

Subsequently, the *tibialis cranialis* muscle, as an effector muscle of the common peroneal nerve, was also approached and evaluated by ultrasound (Figure 2). It was identified subcutaneously as the first muscle mass arising laterally to the crest of the tibia, occupying a craniolateral position in the leg. Muscle width and thickness were determined in triplicate at three different levels: close to its origin, in the middle of its muscle belly, and close to its insertion. The ultrasound characteristics of the muscle and its relationship with neighboring soft tissues were also evaluated.

## 3. Results

### 3.1. Ultrasound Technique

The applied ultrasound technique made it possible to identify the sciatic nerve and its two terminal branches, the common peroneal nerve and the tibial nerve, to assess its ultrasound characteristics and dimensions as well as the topographical and anatomical relationships with neighboring tissues. It also allowed for identifying, evaluating, and sizing the cranial tibial muscle.

#### 3.1.1. Ultrasound Nerve Evaluation

At the level of the greater ischiatic foramen, the sciatic nerve can be seen advancing between the dorsocaudal aspect of the acetabulum and cranial to the ischial tuberosity, passing over the muscle mass constituted by the gemelli and quadratus femoris muscles, close to the piriformis muscle and advancing caudally to the femur. The presence of these bony structures in the region, although they can be used as anatomical landmarks for nerve identification, can make a more proximal visualization difficult due to the curvature of the sciatic nerve and the hyperechogenicity of the bone that decreases contrast (Figure 3). Accompanying the nerve distally, it is visible between the muscles of the thigh, namely medially to the biceps femoris muscle, laterally to the adductor muscles, and caudally to the femur and the vastus lateralis muscle (Figure 4). 

At the level of the stifle, and with great interindividual variation, the two branches of the sciatic nerve, the common peroneal nerve, and the tibial nerve, are easily observable and distinguishable, diverging from the main nerve (Figure 5). Both the sciatic nerve and its branches appear as hypoechogenic neuronal tubular structures (nerve fascicles and the perineurium) surrounded by a hyperechogenic envelope corresponding to the connective tissue nerve wrappings (epineurium) (Figure 3, Figure 4 and Figure 5). The most lateral nerve, appearing dorsally on the ultrasound image, corresponds to the common peroneal nerve and imagiologically has a slightly smaller diameter than the tibial nerve, which is more medial and appears ventrally on the ultrasound image. Accompanying both nerves distally, there is a tendency for a slight decrease in their diameter before the final ramifications, but the intended site for inducing injury to the common peroneal nerve is proximal to the point where the decrease in diameter occurs.

The diameters of the sciatic nerve were measured in the middle of its course along the thigh and immediately before its division. The diameters of the common peroneal and tibial nerves were determined immediately after its diversion (Figure 6). The values obtained while measuring the diameters of the nerves of the 10 evaluated animals can be consulted in Table 1. The diameter of the sciatic nerve does not vary greatly between its course in the middle of the thigh and immediately before its branching site. Confirming the ultrasound observation, the diameter of the tibial nerve is systematically higher than that of the corresponding common peroneal nerve, distally to their branching site.

#### 3.1.2. Ultrasound Muscle Evaluation

Tibialis cranialis muscle width and thickness were determined in triplicate at three different levels: close to its origin, in the middle of its muscle belly, and close to its insertion. The ultrasound appearance of the muscle is speckled due to the reflection of the perimysial connective tissue, which appears moderately echogenic and allows the fascial architecture of the muscle to be seen. The limits of the muscle are clearly visible since the epimysium that covers it is a very reflective structure (Figure 7). The muscle appears as the first muscle mass arising laterally to the crest of the tibia, occupying a craniolateral position in the leg, cranially to the tibia. The thickness and width values measured close to its origin, in the middle of its muscle belly, and close to its insertion can be found in Table 2. As expected, the dimensions of the muscle are greatest in the middle of the belly of the muscle, being smallest close to the insertion site on the tarsus and metatarsus, and presenting intermediate values at their origin in the lateral condyle of the tibia, lateral edge of the tibial tuberosity, and small surface on the lateral surface of the tibia.

## 4. Discussion

Despite all the advances achieved in recent years in the field of regenerative medicine applied to peripheral nerves after injury, there are still many doubts regarding the best and most effective therapeutic approaches to be applied in these clinical scenarios and also about how effective these treatments are in its ability to allow the injured nerve to return to its original form and function.

In human patients and companion animals of clinical interest, the assessment of nerve recovery after injury is largely based on the self-description of symptoms, motor and sensory functional assessment, and electrophysiological studies [17,18]. The imaging approach has been sparsely used and has only recently gained more importance in this area as dramatic advances are observed in the quality of the techniques that can be used. In traditional and smaller animal models such as rodents, some commonly used imaging approaches include techniques such as transgenic labeling, nervous stereology, and muscle histomorphometry [6,11]. In larger animal models, however, although these routine techniques can and should also be used, other methods closer to those likely to be applied in real clinical settings, such as ultrasound, are also an option that should be considered [19].

Ultrasound is an imaging technique routinely used in medical practice that can be used to explore peripheral nerves in all limbs [12], providing information on the consequences of an injury and on the regenerative process in a complementary way to those obtained by electrophysiological techniques and morphological and functional evaluation. This method without radiations allows not only to identify the peripheral nerves among other neighboring tissues in real-time and in high resolution but also to distinguish nerve components such as the *epineurium* and the interfascicular *perineurium* [20]. In this way, the ultrasound will also allow for identifying the presence of structural alterations in the nerve, such as disruption of nerve continuity and the presence of other abnormalities [14,21]. After an injury and during the follow-up of nerve regeneration, ultrasonography will also allow for identifying the fraction of preserved and injured nerve fascicles, the occurrence of postoperative complications such as the development of neuromas, misdirection of nerve growth, exuberant edema, and infiltration of fibrous tissue and/or fat, and also allows procedures such as ultrasound-guided administration of therapies or anesthesia by nerve block [22,23]. Experimentally, after applying a therapeutic approach such as neural guide conduits, ultrasound also makes it possible to monitor the growth of the nerve within the tubular lumen and record the filling of the tube, the reconnection of the nerve tops, and the restoration of anatomical continuity after a neurotmesis injury [24].

Despite this, ultrasound has some disadvantages, starting with the fact that the nerves to be approached must have a superficial position in order to obtain images and details of greater quality. Nerves in a deeper position or very close to hyperechogenic structures, such as bones, make it difficult to obtain quality images with good contrast (Figure 3) [19]. Although it is a cheaper, less time-consuming, more accessible technique, and ultrasound scanners are more portable and easier to use routinely than the machinery of other techniques such as magnetic resonance imaging, ultrasound allows a lower contrast resolution. However, locally, ultrasound allows a better spatial resolution than resonance applied to the whole body [25]. Microscopically, ultrasonography does not allow an assessment of the anatomy and regenerative phenomena occurring at a subfascicular level, such as axonal and endoneurial growth, requiring the complementary use of stereological techniques.

Our research group has been developing and improving more complex animal models to be applied in peripheral nerve regeneration assays, namely the ovine model [9]. This model has several advantages over other options, namely its low acquisition and maintenance costs, docile temperament and easy handling, advanced longevity, and the reduced ethical limitations associated with its use. In addition, sheep have peripheral nerve anatomical and physiological characteristics identical to humans, allowing the induction of lesions to be mimicked and the application of therapeutic options to promote a regenerative process similar to that observed in real clinical scenarios [7]. 

In our previous works, we chose to select the common peroneal nerve as the one to be explored as a model of nerve injury since an injury to the sciatic nerve would bring too severe functional consequences for the sheep, making it impossible to comply with the principles of animal welfare needed to carry out this type of work, namely causing changes in motor capacity, weight support, environmental exploration, and food consumption. Injury to the common peroneal nerve causes functional alterations that are easily identifiable and traceable over time, such as passive flexion of the digits, extension of the tarsus, and loss of sensitivity in the dorsal–distal region of the limb, while allowing the animal to withstand the weight, walking, and carrying out typical environmental exploration and not jeopardizing their well-being. Finally, the common peroneal nerve has a superficial and subcutaneous position in part of its course on the lateral face of the hind limb, allowing for easy surgical access and exploration.

Thus, surgical protocols for inducing injury and application of therapeutic options were previously established, as well as functional assessment techniques that allow monitoring the progression of functional recovery after injury and treatment of the common peroneal nerve. In order to maximize the amount of information obtained in the characterization of the regenerative process of this nerve over time, in this work, an ultrasound evaluation of the common peroneal nerve and other regionally related nerves, the sciatic and the tibial nerves, was carried out, looking for characterizing these nervous structures and their relationship with other neighboring tissues by ultrasound. The *tibialis cranialis* muscle was also explored by ultrasound as an effector muscle of the common peroneal nerve. After this characterization, the dimensional values obtained and the described ultrasound appearance will serve as control values to be compared with those obtained in nerves and muscles subjected to experimental nerve injuries. The superficial position of the nerves and of the *tibialis cranialis* muscle and the housing environment of the sheep make ultrasound the fastest, cheapest, and most effective imaging technique for obtaining the intended information.

The sciatic nerve was easily identified mid-thigh medially to the *biceps femoris* muscle and caudal to the femur and *vastus lateralis* muscle, allowing these muscles to be used as reference points for quick identification even for less experienced ultrasonographers. Monitoring the nerve in a proximal direction makes it possible to identify its passage between the greater trochanter of the femur and the ischial tuberosity, and these bony projections can be used as a reference point for its location. More proximally is its emergence from the greater sciatic foramen. At this level, the nerve may be more difficult to identify due to its curvature and also due to its proximity to the caudal gluteal artery and vein, but the use of a color-flow Doppler may clear up any doubts. Confusion with the sacred loin trunk in this region will be more unlikely since it is at greater depth. The sciatic nerve can also be easily tracked distally to the point where it branches proximally to the stifle. As the nerve progresses, an increase in echogenicity seems to be observed, a phenomenon described in other species [16], and which is probably related to an increase in the connective tissue within the nerve. It is important to remember that this increase in the amount of connective tissue is also related to an increased likelihood of compression or stretching injuries, making it harder to observe the nerve and apply local anesthetic blocks [26]. The mean diameter of the sciatic nerve measured mid-thigh and just before its branching site is identical to that indicated in other species, such as dogs of similar size. Its ultrasound appearance is similar to that described in other species, such as humans [27], dogs [14,15], or cats [16], appearing as a hypoechogenic tubular structure covered by a hyperechogenic envelope corresponding to the *epineurium*, an aspect that allows its easy differentiation from soft tissues and bones in the vicinity. Before its branching site, the sciatic nerve diameter was also measured in a cross-section, rotating the probe 45 degrees from the position that allows for obtaining the longitudinal image (Figure 8). Obtaining an image in this plane proved to be more complex and made it difficult to establish anatomical relationships with neighboring structures and with the ramifications of the sciatic nerve. On the other hand, the ultrasound image reveals characteristics identical to those obtained in the regular plane, in addition to the measured diameter being the same as that obtained with the previous technique. Therefore, it was decided not to use the transverse plane for this ultrasound characterization in any of the considered nerves.

After the branching of the sciatic nerve into its main branches, the common peroneal nerve arises laterally regarding the tibial nerve (dorsally in the ultrasound image), projecting in its course towards the extensor muscles of the fingers and flexors of the tarsus and presenting a systematically smaller diameter than that of the tibial nerve, as described in the literature for other species [16,28]. The latter arises medially (ventrally in the ultrasound image), projecting towards the flexor muscles of the fingers and extensor muscles of the tarsus, with a larger diameter. The diameter of the common peroneal nerve is identical to that previously identified in this species through stereology techniques [9]. It is expected that the diameter of the three nerves may vary according to the size of the animals, but as all the sheep used in this work had identical body dimensions, no significant variations were observed. The ultrasound aspect of the common peroneal and tibial nerves is identical to that of the sciatic nerve, and the difficulty in observing these nerves arises distally, where their diameter can be significantly reduced before their final ramifications. Despite this, this fact should not be a limitation in experimental work since the intended site for inducing lesions in the common peroneal nerve will be shortly after the site of emergence under the *biceps femoris* muscle, where the nerve becomes superficial but still has significant dimensions that will allow easy identification, isolation, induction of lesions and application of therapeutic options [9].

The described technique allows an easy follow-up of the progression of nerve regeneration over time after the induction of a lesion in a controlled surgical environment. For example, after the application of a tube guide, in which the nerve tops of a transected nerve are sutured to the ends of the tube, leaving a gap to be filled inside the biomaterial, immediately after surgery, it is possible to identify the tube guide and the nerve tops inside of it by ultrasound (Figure 9). The tube guide appears as a hyperechogenic structure, at the ends of which the nerve tops are observed (Figure 9a), allowing the measurement of the created gap and its evaluation and measurement over time until the total closure and reconnection of the nerve ends (Figure 9b). In the same way, after the application of end-to-end sutures, in which the ends of the transected nerve are coapted and sutured in order to guarantee their reconnection and anatomical continuity, it is possible to observe the nerve continuity (Figure 10), but at early timepoints after the injury, a small hypoechoic gap appears as a result of the transection. An edema, in association with the inflammatory infiltration resulting from the degenerative phase after the nerve injury, promotes a transient increase in the diameter of the nerve (Figure 10a) that ends up disappearing over time (Figure 10b). In addition to the increase in diameter, due to interstitial edema, the nerve also appears with an increased hyperechogenic appearance. The main advantage of applying ultrasound in the monitoring of nerve regeneration will be to allow the observation of macroscopic morphological changes in the nerve over time, namely changes in its anatomical continuity, in the dimensions of the created nerve gaps, in their diameter, and in the presence or absence of edema and inflammatory infiltrate. Microscopic changes such as endoneurial microvascular degeneration, demyelination/remyelination, and axonal density and reorganization should be later confirmed by more sensitive histomorphometric techniques such as nerve stereology. At the same time, the functional translation of the regenerative process can be evaluated in parallel through functional and behavioral tests, determination of nervous conductivity, and biomechanical and kinematic gait studies.

Among the muscles innervated by the common peroneal nerve is the cranial tibial muscle, a muscle with functions of flexion of the tarsus, extension of the digits, and flexion of the stifle. Due to its anatomical position, this muscle is easily identifiable in a craniolateral position regarding the tibia and can be accessed both by palpation and ultrasound. When a common peroneal nerve injury occurs, the *tibialis cranialis* muscle is among those that may suffer atrophy by denervation, and its collection for histomorphometry analysis is common to compare the level of histological reorganization of the muscle with the quality of nerve regeneration [5]. In this work, the *cranialis tibialis* muscle of sheep was also evaluated for characterization, description of its ultrasound image, and determination of normal dimensions in animals not subject to injury. The muscle was easily identified by ultrasound, extending from the condyles of the tibia to the tarsus/metatarsus along the lateral aspect of the leg. The muscle appears delimited by the hyperechogenic *epimysium* that differentiates it from other extensor muscles in the vicinity and also from the tibia more deeply. The muscle mass is speckled due to the presence of echogenic connective tissue from the perimysium between the nerve fascicles. The two main dimensions of muscle, width and thickness, were measured in three different positions to establish normal values for healthy muscles. These data will be important in future work to compare muscle characteristics in healthy animals with those subject to denervation atrophy after injury of the common peroneal nerve. In this situation, the muscle will tend to lose muscle mass, appearing with reduced dimensions, and may also appear hyperechogenic due to phenomena of fibrosis, inflammation, and fatty infiltration [8,29].

Despite the success in establishing the ultrasound technique described in this work, it is not free from some limitations that must be considered. The sheep used to obtain ultrasound images were healthy animals of similar ages that were purchased from authorized national producers previously approved by the host institution. However, they were adult animals not bred specifically for use in animal experimentation, which means that they had a varying clinical history and showed noticeable dimensional and anatomical variations. This translates into an obvious variability in the image and dimensions recorded, even with the care that was contemplated to reduce interoperator variability. Those variations must also be considered in other works where the observation of ultrasound images and structural dimensions different from those indicated here can happen, especially with the use of animals of other breeds or in a different age range. Although the number of 10 animals may be considered adequate to establish reliable reference values as a starting point for future work, a larger number of records will be advantageous, increasing the number of animals used to establish control values for the dimensions of peripheral nerves and effector muscles. This can be achieved by evaluating additional healthy animals in future trials, for example, using healthy limbs considered as controls [9]. Another limitation associated with the technique applied here is related to the fact that the animals used during the ultrasound recordings were subjected to prior sedation to ensure the safety of the procedures, animal welfare, and maintenance of the material used. This preparation makes the process time-consuming, particularly considering its future application in preclinical trials where to guarantee a correct characterization of the ultrasound image of the nerves subject to injury and during their regenerative process, evaluations will be necessary at several timepoints. Even so, this prior procedure is essential, considering the fearful nature of sheep, which do not easily tolerate correct handling when lying down for long minutes or in the presence of several operators in the vicinity. This limitation will no longer exist when, in the future, these techniques will be clinically applied to other species of medical and clinical interest, such as companion animals or horses.

## 5. Conclusions

The imaging approach described in this work allows the use of ultrasonography as an additional diagnostic tool applied in the evaluation of the peripheral nerves of large animal models. In this way, it will be possible to evaluate the common peroneal nerve subject to an experimentally induced injury and confirm nerve regeneration over time, comparing the variations of the ultrasound image with the evolution observed in other nerve and muscle functional evaluation parameters. In the future, the ultrasound images and the values obtained in this work may be used as control parameters in preclinical trials testing new therapeutic approaches in the field of PNI. 

## Figures and Tables

**Figure 1 life-13-01919-f001:**
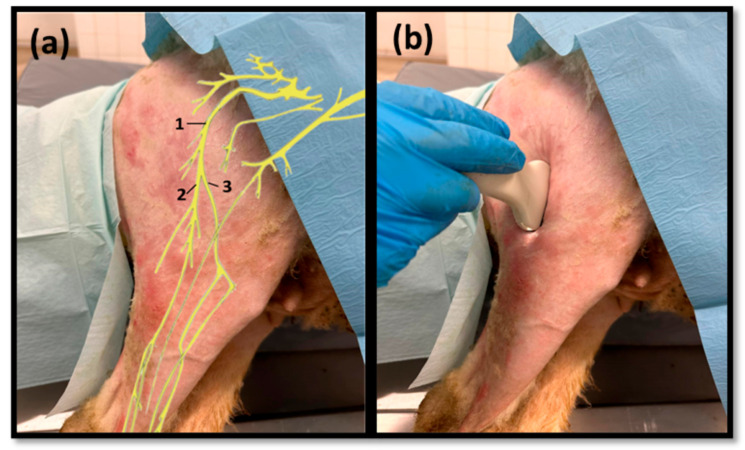
(**a**) Schematic representation of the distribution of the sciatic nerve (**1**) and its branches and tibial (**2**) and common peroneal (**3**) nerves in the hind limb; (**b**) demonstration of the applied ultrasound technique.

**Figure 2 life-13-01919-f002:**
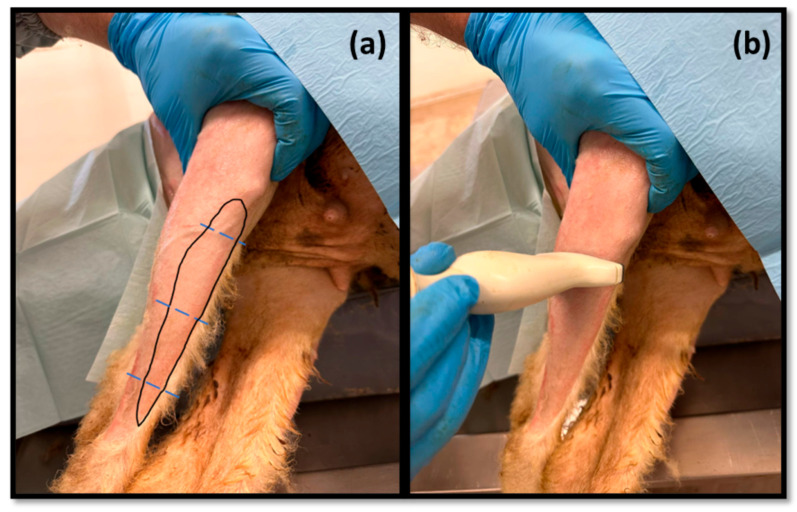
(**a**) Schematic representation of the location of the cranial tibial muscle on the cranial surface of the leg; (**b**) demonstration of the applied ultrasound technique. The blue dashed lines indicate the measurement sites.

**Figure 3 life-13-01919-f003:**
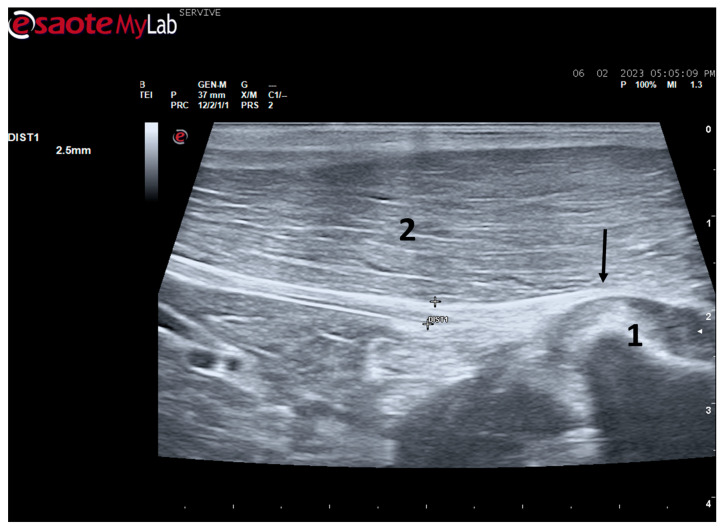
Ultrasound image of the left sciatic nerve of a sheep after emergence from the greater sciatic foramen: (**1**) greater trochanter of the femur; (**2**) biceps femoris muscle; arrow—curvature of the sciatic nerve when passing between the reference bone structures. DIST1 represents the measurement of the diameter of the sciatic nerve.

**Figure 4 life-13-01919-f004:**
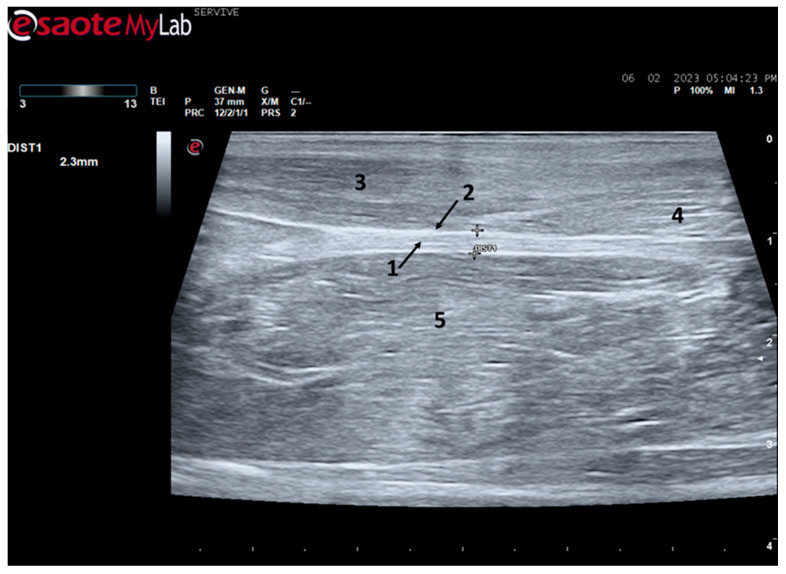
Ultrasound image of the left sciatic nerve of a sheep nerve halfway through the thigh: (**1**) sciatic nerve, identified by an arrow (nerve fascicles enclosed by perineurium); (**2**) epineurium, identified by an arrow; (**3**) biceps femoris muscle; (**4**) piriformis muscle; (**5**) adductor muscles. DIST1 represents the measurement of the diameter of the sciatic nerve halfway through the thigh.

**Figure 5 life-13-01919-f005:**
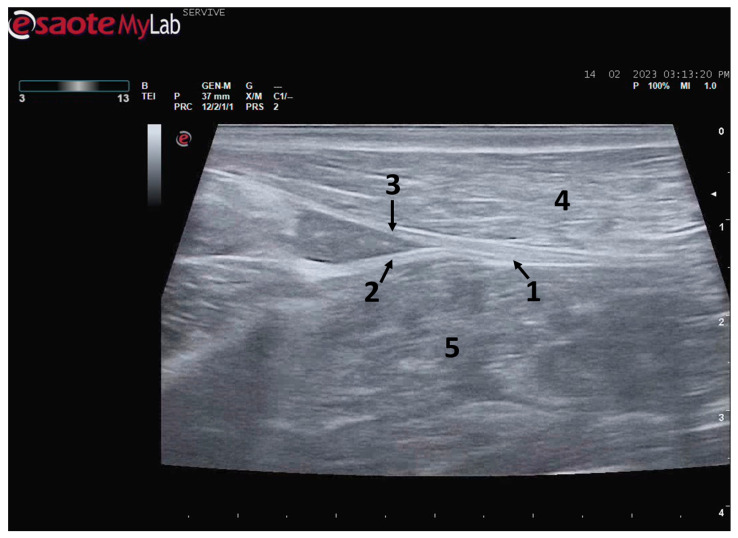
Ultrasound image of the left sciatic nerve and its ramifications in a sheep: (**1**) sciatic nerve, identified by an arrow; (**2**) tibial nerve, identified by an arrow; (**3**) common peroneal nerve, identified by an arrow; (**4**) biceps femoris muscle; (**5**) adductor muscles.

**Figure 6 life-13-01919-f006:**
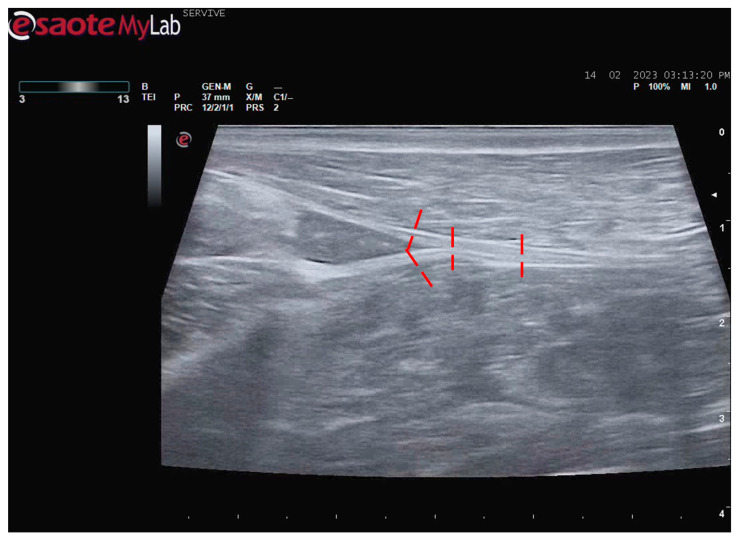
Schematic representation of the measurement sites for the diameter of the sciatic, common peroneal, and tibial nerves. The measurement sites are identified by the red dashed lines.

**Figure 7 life-13-01919-f007:**
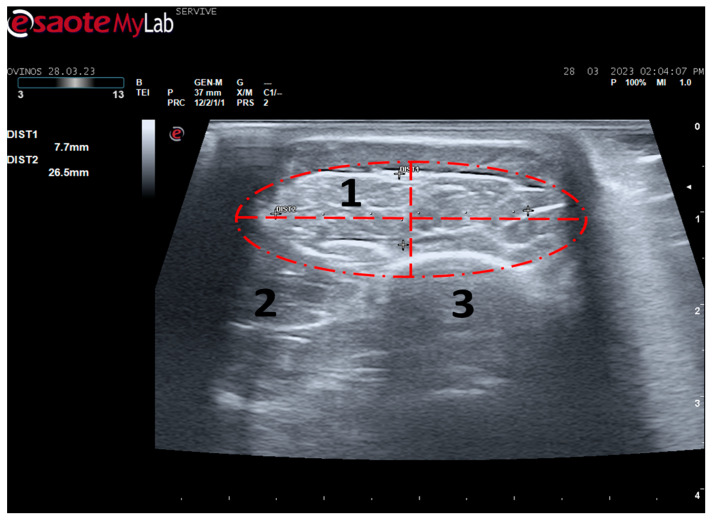
Ultrasound image of the left cranial tibial muscle taken close to its origin, bounded by the red dashed circle. (**1**) Tibialis cranialis muscle; (**2**) extensor digitorum longus muscle; (**3**) tibia. DIST1 (vertical red dashed line) and DIST2 (horizontal red dashed line) represent the thickness and width of the muscle, respectively.

**Figure 8 life-13-01919-f008:**
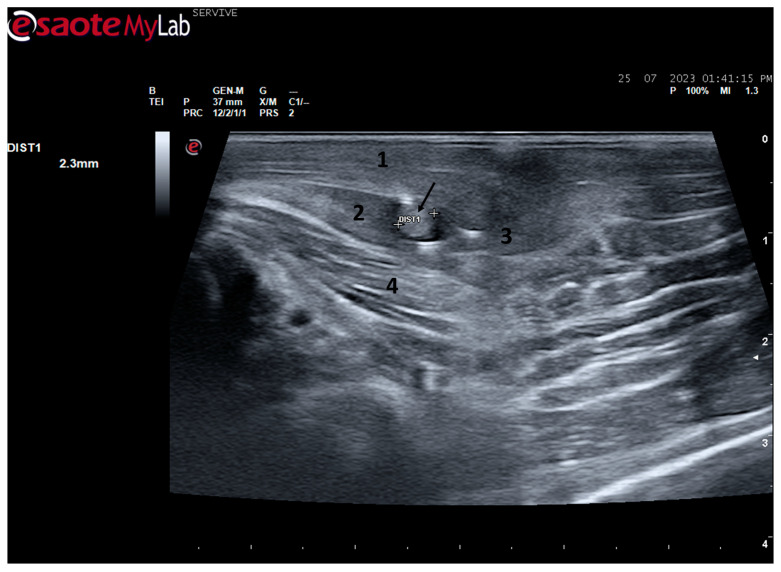
Ultrasound image of the left sciatic nerve before its branching site, cross section (black arrow). In this plane, the nerve appears as an echogenic tubular structure surrounded by the musculature of the region. (**1**) Biceps femoris muscle; (**2**) vastus lateralis muscle; (**3**) Semitendinosus muscle (**4**) adductor muscles. DIST1 represents the measurement of the diameter of the sciatic nerve before its branching site.

**Figure 9 life-13-01919-f009:**
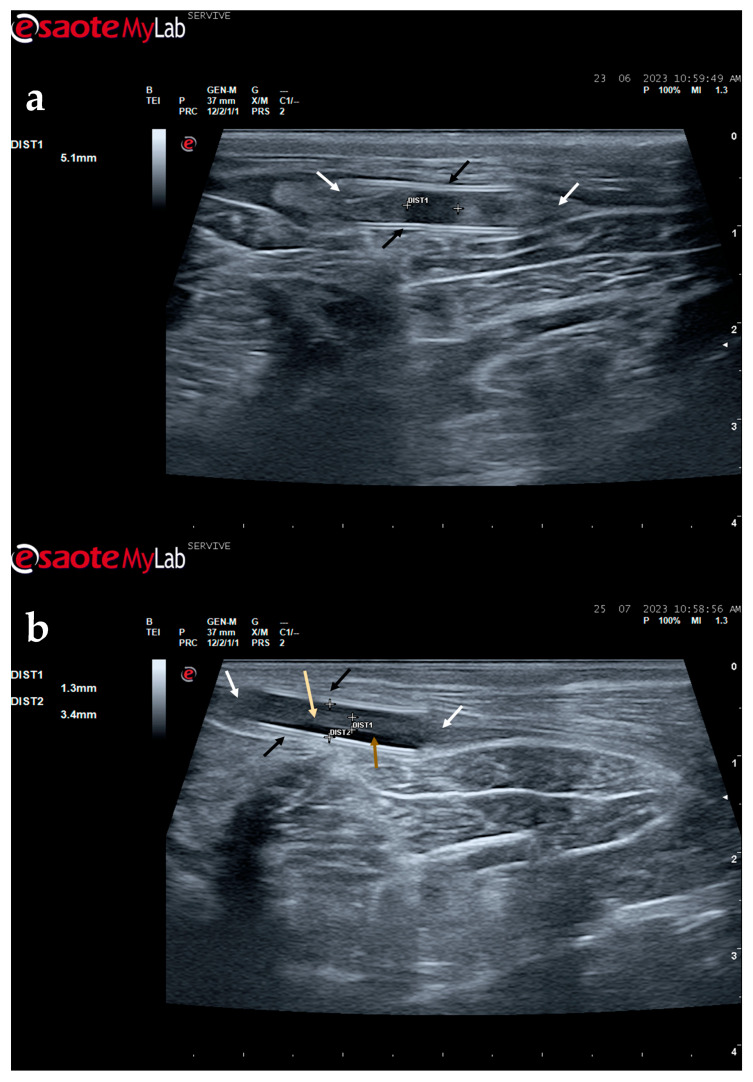
Ultrasound image of the left peroneal nerve in a sheep subjected to a transection lesion followed by the application of a tube guide. (**a**) After 1 week, it is possible to observe the tube guide as a hyperechogenic tubular structure (black arrows) at the ends of which the ends of the transected nerve are inserted (white arrows). DIST1 represents the length of the gap left between the two nerve ends. (**b**) After 3 months, the tube continues to be perfectly visible (black arrows), and the nerve is introduced into its lumen (white arrows); but now, instead of the gap between the nerve tops, an anatomical continuity of the nerve is observed along the entire tubular lumen (beige arrow), indicating a nerve reconnection. In the center of the tubular lumen, a hypoechoic space not filled by nervous tissue is also observed (brown arrow), indicating that the regenerating nerve has not yet occupied all the available space inside the tube guide. DIST1 represents the nerve diameter, and DIST2 represents the inner diameter of the tube guide.

**Figure 10 life-13-01919-f010:**
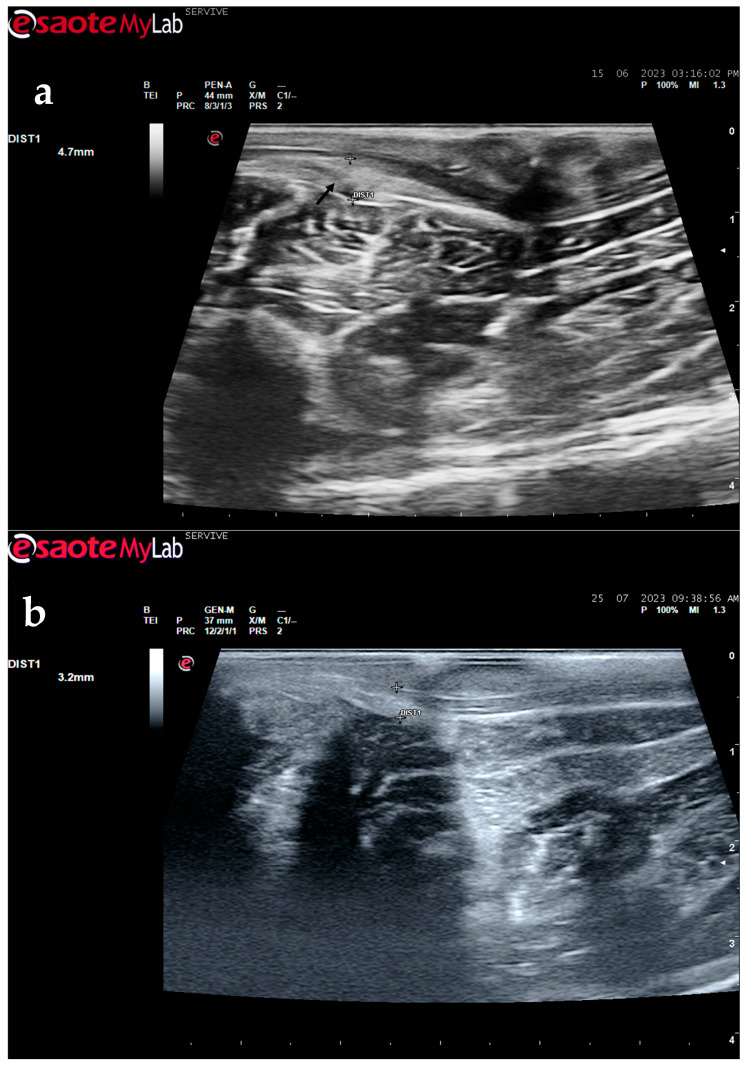
Ultrasound image of the left peroneal nerve in a sheep subjected to a transection lesion followed by the application of an end-to-end suture. (**a**) After 1 week, it is possible to observe that the nerve continuity is still interrupted by a gap between the two nerve tops that have not yet fully reconnected (black arrow). DIST1 represents the nerve diameter, which is enlarged and with an increased hyperechogenic appearance due to interstitial edema and inflammatory infiltrate. (**b**) After 3 months, the nerve already presents an anatomical continuity, it is less hyperechogenic, and its diameter has decreased considerably (DIST1).

**Table 1 life-13-01919-t001:** Diameter values measured in the sciatic nerve at the mid-thigh level, at the bifurcation site, and in the common peroneal and tibial nerves immediately after branching. Values expressed as mean and standard deviation (SD). n = 10.

			Mean	SD
Sciatic n.	Middle thigh	Left	2.32	0.03
		Right	2.29	0.01
	Branching site	Left	2.28	0.09
		Right	2.27	0.02
Branches	Common Peroneal n.	Left	1.36	0.05
		Right	1.37	0.01
	Tibial n.	Left	1.92	0.03
		Right	1.91	0.01

**Table 2 life-13-01919-t002:** Thickness and width values (mm) obtained from the cranial tibial muscle at the level of its origin, middle belly, and insertion. Values expressed as mean and SD. n = 10.

				Mean	SD
Cranial tibial muscle	Origin	Left	Thickness	6.50	0.04
			Width	20.33	0.02
		Right	Thickness	6.50	0.02
			Width	20.01	0.08
	Middle belly	Left	Thickness	7.61	0.03
			Width	18.74	0.03
		Right	Thickness	7.42	0.15
			Width	18.62	0.03
	Insertion	Left	Thickness	5.79	0.05
			Width	16.07	0.06
		Right	Thickness	5.76	0.03
			Width	16.05	0.02

## Data Availability

The data that support the findings of this study are available from the corresponding author on request.

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
