# Peer review of "Ultrasound Landmarks in the Approach to the Common Peroneal Nerve in a Sheep Model—Application in Peripheral Nerve Regeneration"

_life, 2023, doi:10.3390/life13091919_

Round 1

Reviewer 1 Report (Previous Reviewer 2)

The authors convincingly answered the reviewers' questions and corrected them according to the comments. Manuscripts in their present form may be accepted for publication.

 Minor editing of English language required.

Author Response

Answer to reviewer 1

Dear reviewer 1:

The authors are grateful for the feedback given by the reviewer and acceptance of the article in its current form. We believe that the improvements that were introduced in the previous phase following the reviewer's suggestions have contributed to a significant improvement in the overall quality of the article.

Reviewer 2 Report (New Reviewer)

In the present study, Rui Alfites et al. investigated the potential use of ultrasound in peripheral nerve regeneration using a sheep model. The study is titled "Echographic Landmarks in the Approach to the Common Peroneal Nerve in a Sheep Model: Implications for Peripheral Nerve Regeneration." However, there are several aspects that require clarification by the authors:

1. What is lacking in the present study is the utilization of animals with experimentally induced peripheral injuries. All animals utilized in the study were sheep with the same baseline condition. There is no comparison with a disease model or animals that have actually experienced peripheral injury.

2. The authors investigate the use of ultrasound in assessing peripheral nerves in sheep, thereby proposing it as a potential tool for investigating nerve injuries. However, the title they have chosen appears to make an overly ambitious claim: "Implications for Peripheral Nerve Regeneration."

3. In the discussion section, the authors briefly mentioned limitations associated with ultrasound use. However, they did not mention a comprehensive discussion of the limitations inherent in the present study, including experimental design and other relevant factors.

Author Response

Answer to reviewer 2

Dear reviewer 2:

Thank you very much for the feedback on this review phase, and also for the suggestions made, which received the best attention from us. The authors inform that the final document has been revised and all suggestions made by the reviewers have been introduced, being duly identified in the final document with highlight in different colors. This review also made it possible to identify and correct some errors and typos as well as improve the general level of English.

  1. What is lacking in the present study is the utilization of animals with experimentally induced peripheral injuries. All animals utilized in the study were sheep with the same baseline condition. There is no comparison with a disease model or animals that have actually experienced peripheral injury.

The authors would like to reinforce that the aim of this article is not to apply the principles of ultrasound in a specific work on peripheral nerve regeneration, but rather to establish the anatomical principles and the ultrasound technique that will allow the approach of the peripheral nerves considered in future works. It is, as such, a guide, which not only describes in detail the step-by-step process necessary for the application of this technique in future works, but also records standard images of healthy peripheral nerves and tibialis cranialis muscles and measurements taken on these structures, to be used as controls in future works. These indications are effectively present in the submitted article, namely in the following passages, which were highlighted in yellow in the final document to facilitate their identification:

“The results obtained will allow the use of these data as control values in future work exploring new therapeutic options for nerve regeneration after induction of common peroneal nerve injuries in sheep.”

“In this work, an ultrasound evaluation of ten sheep’s sciatic nerves and its main branches, the common peroneal nerve, and the tibial nerve, was carried out in order to establish the echographic appearance of these structures in non-intervened nerves and its anatomical and regional relationship to neighboring structures.”

“Subsequently, the registered features may be used as reference values in comparative studies with peroneal common nerves subjected to controlled injuries and with cranial tibial muscles that have suffered a consequent atrophy due to denervation, maximizing the number of parameters to be evaluated in the intervened animals and ensuring a better determination of the performance of the applied therapeutic options.”

“In the future, the ultrasound images and the values obtained in this work may be used as control parameters in pre-clinical trials testing new therapeutic approaches in the field of PNI.”

In this way, considering that it was directly established in the article that its aim is to work as a previous step to the application of the techniques described in future specific works, the authors consider that in its present form the document fulfills the objectives that it proposes. Effectively, the authors are currently running a project in which the assumptions described in this article are being applied in real scenarios of peripheral nerve regeneration, and whose results will be published in the future with reference to the present article as an introduction and methodological description.

  1. The authors investigate the use of ultrasound in assessing peripheral nerves in sheep, thereby proposing it as a potential tool for investigating nerve injuries. However, the title they have chosen appears to make an overly ambitious claim: "Implications for Peripheral Nerve Regeneration."

The authors would like to draw attention to the fact that the title chosen for the article is "application in peripheral nerve regeneration" instead of "Implications for Peripheral Nerve Regeneration" as indicated by the reviewer. The justification for choosing this title lies in the response to the previous point and is essentially linked to the authors' objective of establishing this article as an introduction to the application of the technique described here in regenerative medicine works in sheep that are currently taking place, and that will be published in the future with reference to this article.

  1. In the discussion section, the authors briefly mentioned limitations associated with ultrasound use. However, they did not mention a comprehensive discussion of the limitations inherent in the present study, including experimental design and other relevant factors.

The following text segment has been inserted, and appears in the final document highlighted in yellow:

“Despite the success in establishing the ultrasound technique described in this work, it is not free from some limitations that must be considered. The sheep used to obtain ultrasound images were healthy animals of similar ages that were purchased from authorized national producers previously approved by the host institution. However, they were adult animals not bred specifically for use in animal experimentation, which means that they had a varying clinical history and showed noticeable dimensional and anatomical variations. This translates into an obvious variability in the image and dimensions recorded, even with the care that was contemplated to reduce interoperator variability. Those variations must also be considered in other works where the observation of ultrasound images and structural dimensions different from those indicated here can happen, especially with the use of animals of other breeds or in a different age range. Although the number of 10 animals may be considered adequate to establish reliable reference values as a starting point for future work, a larger number of records will be advantageous, increasing the number of animals used to establish control values for the dimensions of peripheral nerves and effector muscles. This can be achieved by evaluating additional healthy animals in future trials, for example using healthy limbs considered as controls (9). Another limitation associated with the technique applied here is related to the fact that the animals used during the ultrasound recordings were subjected to prior sedation to ensure the safety of the procedures, animal welfare and maintenance of the material used. This preparation makes the process time-consuming, particularly considering its future application in pre-clinical trials where, to guarantee a correct characterization of the ultrasound image of the nerves subject to injury and during their regenerative process, evaluations will be necessary at several timepoints. Even so, this prior procedure is essential, considering the fearful nature of sheep, which do not easily tolerate correct handling when lying down for long minutes or the presence of several operators in the vicinity. This limitation will no longer exist when, in the future, these techniques will be clinically applied to other species of medical and clinical interest, such as companion animals or horses.”

Reviewer 3 Report (New Reviewer)

The authors subjected an ultrasonographic study of sheep's three nerves in the hindlimb and the cranial tibial muscle to obtain the control values for future work exploring new therapeutic options for nerve regeneration. The paper has, in general, the informative value.

The idea itself is promising but the paper includes some flaws which should be corrected or explained.

1. Apart from the editing side of the work, which deviates from MDPI standards, especially the references, which are incorrect (or even missing in refs. 5, 26), the title does not fully refer to the methodology used in the work: "Echographic landmarks in the approach to the common peroneal nerve"... . The Authors used ultrasonographic study of three peripheral nerves of the hind limb, the common peroneal, tibial, and sciatic nerves, and an evaluation of the cranial tibial muscle was performed. I suggest deleting from the text the synonym “echographic” and using only the term "ultrasonography", to more appropriate reflect the whole scope of the paper.

2. The Abstract does not include a clearly precise aim, but the general description of what the Authors intended to achieve in their study.

3. This is not the paper dealing directly with the regeneration, but the first two keywords are directly related to this issue.  

4. In the Introduction, please clarify the sentence ..."it has not yet been possible to establish an unequivocal alternative treatment to supplant traditional surgical methods as gold standard approaches (2)."... It sounds confusing. Similarly, please rewrite …” Most  studies  of  ultrasonographic  description  of  peripheral  nerves  in  Veterinary Medicine were carried out in companion (?) animals””… . 

Please write one sentence of clear, short precise aim in lines 110-111.

5. In the Results section, there are presented data on measurements from ultrasonography studies. They are not commented on in the Discussion in comparison to other, similar anatomical works in a sheep. Do they really not exist?

Minor corrections required

Author Response

Answer to reviewer 3

Dear reviewer 3:

Thank you very much for the feedback on this review phase, and also for the suggestions made, which received the best attention from us. The authors inform that the final document has been revised and all suggestions made by the reviewers have been introduced, being duly identified in the final document with highlight in different colors. This review also made it possible to identify and correct some errors and typos as well as improve the general level of English.

  1. Apart from the editing side of the work, which deviates from MDPI standards, especially the references, which are incorrect (or even missing in refs. 5, 26), the title does not fully refer to the methodology used in the work: "Echographic landmarks in the approach to the common peroneal nerve"... . The Authors used ultrasonographic study of three peripheral nerves of the hind limb, the common peroneal, tibial, and sciatic nerves, and an evaluation of the cranial tibial muscle was performed. I suggest deleting from the text the synonym “echographic” and using only the term "ultrasonography", to more appropriate reflect the whole scope of the paper.

The term "echography" was replaced throughout the text by the term "ultrasound", including in the title of the article, and the necessary textual adaptations were made for this change. New terms appear highlighted in green for better identification.

References 5 and 26 were confirmed and are both integrated in the text and in the list of references (highlighted in green).

  1. The Abstract does not include a clearly precise aim, but the general description of what the Authors intended to achieve in their study.

The following passage in the abstract was slightly modified to create a sentence more indicative of the main objective of the work described, appearing in the text highlighted in green:

“In this work, sheep, a promising animal model in the area of peripheral nerve regeneration, were subjected to an ultrasonographic study of three peripheral nerves of the hind limb, the common peroneal, tibial and sciatic nerves. The main aim was to establish values of dimensions and ultrasound appearance in healthy nerves and landmarks for their identification, as well as to perform an ultrasound evaluation of the cranial tibial muscle, an effector muscle of the common peroneal nerve, and to establishing normal values for its ultrasound appearance and dimensions.”

  1. This is not the paper dealing directly with the regeneration, but the first two keywords are directly related to this issue.

The list of keywords has been modified to the following order (highlighted in green):

“Keywords: Peripheral Nerve; Peroneal Common Nerve, Ultrasonography; Sheep Model; Peripheral Nerve Injury; Peripheral Nerve Regeneration.”

  1. In the Introduction, please clarify the sentence ..."it has not yet been possible to establish an unequivocal alternative treatment to supplant traditional surgical methods as gold standard approaches (2)."... It sounds confusing. Similarly, please rewrite …

The sentence was replaced by the following, in order to improve its meaning (highlighted in green):

“…traditional surgical methods continue to be the gold standard treatments commonly used, and therapeutic alternatives with equal effectiveness have not yet been established”.

” Most  studies  of  ultrasonographic  description  of  peripheral  nerves  in  Veterinary Medicine were carried out in companion (?) animals””… .

The sentence was replaced by the following, in order to improve its meaning (highlighted in green):

“Most studies of ultrasonographic description of peripheral nerves in Veterinary Medicine were carried out in smalls animals such as dogs and cats”

Please write one sentence of clear, short precise aim in lines 110-111.

The following sentence has been introduced (highlighted in green):

“The main objective is to establish an ultrasound approach technique and standard values that can be used as reference in future assays on peripheral nerve regeneration in the sheep model.”

  1. In the Results section, there are presented data on measurements from ultrasonography studies. They are not commented on in the Discussion in comparison to other, similar anatomical works in a sheep. Do they really not exist?

As far as the authors are aware, there are no other studies that have established reference values for the dimensions of these peripheral nerves in sheep. As mentioned in the article, the sheep model has not yet been widely explored in this field of research, and the few studies that have been carried out have not focused on this type of approach. This is precisely where the innovation of this work lies.

Therefore, the only comparisons of dimensions were made with other species of similar size, such as dogs, and also with dimensions obtained in sheep by our group in previous works, by stereology, as indicated in the following passages (highlighted in green):

“The diameter of the common peroneal nerve is identical to that previously identified in this species through stereology techniques.”

“The mean diameter of the sciatic nerve measured mid-thigh and just before its branching site is identical to that indicated in other species, such as dogs of similar size.”

Round 2

Reviewer 2 Report (New Reviewer)

All the comments have been clarified and addressed by the authors. Now this article is suitable for publication. 

Author Response

Dear reviewer

The authors acknowledge the revision, best regards 

Reviewer 3 Report (New Reviewer)

I am not still convinced that the ref. Carey B, Barrington M. Nerve Injury in Regional Anaesthesia. is a reliable and PubMed-included source of scientific (but the tutorial ) knowledge. Couldn't it be replaced by some other ref.  with all principles of the PubMed or Scopus citation?

Please complete the ref. Alvites, R. D., Branquinho, M. V., Sousa, A. C., Amorim, I., Magalhães, R., João, F., Almeida, D., Amado, S., Prada, J., Pires, I., Zen, F., Raimondo, S., Luís, A. L., Geuna, S., Varejão, A. S. P., & Maurício, A. C. (2021). Combined Use of Chitosan and Olfactory Mucosa Mesenchymal Stem/Stromal Cells to Promote Peripheral Nerve Regeneration In Vivo. Stem cells international, 2021, 6613029. https://doi.org/10.1155/2021/6613029

My intention in a  query: …” This is not the paper dealing directly with the regeneration, but the first two keywords are directly related to this issue.”…

was to delete the unnecessary keywords not directly related to the peripheral nerve regeneration issue which is not the aim of the study in the discussed  paper. Contrary, the Authors modified keywords to Keywords: Peripheral Nerve; Peroneal Common Nerve, Ultrasonography; Sheep Model; Peripheral Nerve Injury; Peripheral Nerve Regeneration. Please correct with introducing the keywords related to the methodology, animal and anatomical structures of your study.

The Authors still do not follow the rules of the MDPI citation in the list of the refs. the same as in the text.

The authors corrected most of the English language mistakes

Author Response

Answer to reviewer 3

Dear reviewer 3:

Thank you very much for the feedback on this review phase, and also for the suggestions made, which received the best attention from us. The authors inform that the final document has been revised and all suggestions made by the reviewers have been introduced, being duly identified in the final document with highlight in different colors. This review also made it possible to identify and correct some errors and typos as well as improve the general level of English.

  1. I am not still convinced that the ref. Carey B, Barrington M. Nerve Injury in Regional Anaesthesia. is a reliable and PubMed-included source of scientific (but the tutorial ) knowledge. Couldn't it be replaced by some other ref. with all principles of the PubMed or Scopus citation?

The reference in question has been replaced by the following (highlighted in green):

26.O'Flaherty, D.; McCartney, C.J.L., Ng, S.C. Nerve injury after peripheral nerve blockade-current understanding and guidelines. BJA Educ 2018, 18(12), 384-90. https://doi.org/10.1016/j.bjae.2018.09.004.”

  1. Please complete the ref. Alvites, R. D., Branquinho, M. V., Sousa, A. C., Amorim, I., Magalhães, R., João, F., Almeida, D., Amado, S., Prada, J., Pires, I., Zen, F., Raimondo, S., Luís, A. L., Geuna, S., Varejão, A. S. P., & Maurício, A. C. (2021). Combined Use of Chitosan and Olfactory Mucosa Mesenchymal Stem/Stromal Cells to Promote Peripheral Nerve Regeneration In Vivo. Stem cells international, 2021, 6613029. https://doi.org/10.1155/2021/6613029

The reference has been updated, as requested (highlighted in green)

“Alvites, R.D.; Branquinho, M.V.; Sousa, A.C.; Amorim, I.; Magalhaes, R.; Joao, F.; Almeida, D.; Amado, S.; Prada, J.; Pires, I.; Zen, F.; Raimondo, S.; Luis, A.L.; Geuna, S.; Varejao, A.S.P., Mauricio, A.C. Combined Use of Chitosan and Olfactory Mucosa Mesenchymal Stem/Stromal Cells to Promote Peripheral Nerve Regeneration In Vivo. Stem Cells Int 2021, 2021, 6613029. https://doi.org/10.1155/2021/6613029.”

  1. My intention in a query: …” This is not the paper dealing directly with the regeneration, but the first two keywords are directly related to this issue.”… was to delete the unnecessary keywords not directly related to the peripheral nerve regeneration issue which is not the aim of the study in the discussed  Contrary, the Authors modified keywords to Keywords: Peripheral Nerve; Peroneal Common Nerve, Ultrasonography; Sheep Model; Peripheral Nerve Injury; Peripheral Nerve Regeneration. Please correct with introducing the keywords related to the methodology, animal and anatomical structures of your study.

                     As requested, the keywords were changed to the following list, (highlighted in green), eliminating direct references to peripheral nerve regeneration:

                     “Keywords: Peripheral Nerve; Peroneal Common Nerve, Ultrasonography; Ultrasound Landmarks Sheep Model.”

  1. The Authors still do not follow the rules of the MDPI citation in the list of the refs. the same as in the text.

The reference style was modified and adapted to the instructions indicated on the "Instructions for Authors" page, namely by changing the references in the text to square brackets and using the following format as a reference style. DOI was also included:

  1. Author 1, A.B.; Author 2, C.D. Title of the article. Abbreviated Journal Name Year, Volume, page range. DOI.”

This manuscript is a resubmission of an earlier submission. The following is a list of the peer review reports and author responses from that submission.

Round 1

Reviewer 1 Report

The authors have studied the course and diameter of peroneal nerve and related muscles by echography. 

The paper contains findings of the safety on echographi sleep models for assessment of diameter and location of normal peroneal nerves with tables tabulating their results. Although the application for assessment of peroneal nerve regeneration is fascinating, nevertheless it has not been studied in this paper. So the paper needs to be re-written focusing on normal position and diameter of sheep peroneal nerve.    

English is moderate . Please use the  instead of de .

Reviewer 2 Report

The work presents the results of morphological studies, ultrasound of healthy peripheral nerves of sheep. The authors give morphological norms for sciatic, tibial and peroneal nerves, which could supposed be helpful in determining the phase of recovery. Of course, ultrasound will be a complementary method here, because the most important is the physiological measurement of NCV allowing to examine the recovery of nerve conduction function. Therefore, it is equally important to show the NCV results along with the presented morphological values of ultrasound of a healthy nerve.

On the other hand, it would also be important to show an ultrasound image of the damaged nerves of the mentioned group of animals. Can changes such as macrophage infiltration, endoneural microvascular degeneration and demyelination be confirmed by changes in ultrasound, so that ultrasound is a reliable complementary method? At least an explanatory paragraph in the discussion is necessary here.

The work is well and understandably written and the presented results are important for comparative morphology of peripheral nerves.

 Minor editing of English language required.